# Screening of the Pandemic Response Box identifies anti-microsporidia compounds

Qingyuan Huang[1,2], Jie Chen[1], Guoqing Pan[1]*, Aaron W. Reinke[2]*

**1** State Key Laboratory of Resource Insects, Chongqing Key Laboratory of Microsporidia Infection and Control, Southwest University, Chongqing, China, **2** Department of Molecular Genetics, University of Toronto, Toronto, Ontario, Canada

* gqpan@swu.edu.cn (GP); aaron.reinke@utoronto.ca (AWR)

**Data Availability Statement:** The authors confirm that all data underlying the findings are fully available without restriction. All relevant data are within the paper and its Supporting Information file named "S1 Data".

## Abstract

Microsporidia are fungal obligate intracellular pathogens, which infect most animals and cause microsporidiosis. Despite the serious threat that microsporidia pose to humans and agricultural animals, few drugs are available for the treatment and control of microsporidia. To identify novel inhibitors, we took advantage of the model organism *Caenorhabditis elegans* infected with its natural microsporidian *Nematocida parisii*. We used this system to screen the Pandemic Response Box, a collection of 400 diverse compounds with known antimicrobial activity. After testing these compounds in a 96-well format at high (100 μM) and low (40 μM) concentrations, we identified four inhibitors that restored the ability of *C. elegans* to produce progeny in the presence of *N. parisii*. All four compounds reduced the pathogen load of both *N. parisii* and *Pancytospora epiphaga*, a *C. elegans*-infecting microsporidia related to human-infecting species. One of these compounds, a known inhibitor of a viral protease, MMV1006203, inhibited invasion and prevented the firing of spores. A bisindole derivative, MMV1593539, decreased spore viability. An albendazole analog, MMV1782387, inhibited proliferation of *N. parisii*. We tested albendazole as well as 5 other analogs and observed that MMV1782387 was amongst the strongest inhibitors of *N. parisii* and displayed the least host toxicity. Our study further demonstrates the effectiveness of the *C. elegans*-*N. parisii* system for discovering microsporidia inhibitors and the compounds we identified provide potential scaffolds for anti-microsporidia drug development.

## Author summary

Microsporidia are a large group of parasites which infect both humans and agriculturally important animals such as honey bees and shrimp. These parasites pose a threat to health and food security and there are few therapeutic options available to treat them. To identify compounds that could inhibit microsporidia, we took advantage of the roundworm model organism *Caenorhabditis elegans*, and the microsporidia *Nematocida parisii* which naturally infects this worm. We then used this system to screen the Pandemic Response Box, which is a collection of compounds with activity towards pathogens. We identified four compounds which could inhibit microsporidia, and found that these compounds worked by either inactivating microsporidia spores, preventing microsporidia

**Funding:** This work was supported by Canadian Institutes of Health Research grant (no. 461807 to AWR, https://cihr-irsc.gc.ca/e/193.html) and QH was supported by an award from the China Scholarship Council (https://www.chinesescholarshipcouncil.com/). The funders had no role in study design, data collection and analysis, decision to publish, or preparation of the manuscript.

**Competing interests:** The authors have declared that no competing interests exist.

germination, or hindering the growth of the parasite. Together this work provides compounds that could be used to study microsporidia infection and that could be a starting point for the development of drugs against these parasites.

## Introduction

Microsporidia are obligate intracellular pathogens phylogenetically related to fungi [1,2]. Approximately 1700 species of microsporidia have been identified to infect invertebrates, vertebrates, and protists, with at least 17 species known to infect humans [3–5]. In addition to posing a serious threat to human health, microsporidia are responsible for substantial economic losses in agriculture. Species of farmed penaeid shrimp are susceptible to hepatopancreatic microsporidiosis, which is caused by *Ecytonucleospora hepatopenaei* [3]. *Vairimorpha ceranae* and *Vairimorpha apis* are a threat to the global beekeeping industry [6,7]. A lethal pathogen, *Nosema bombycis*, causes heavy losses or even total culture failure in the silkworm industry [8]. Moreover, some species, such as *Ameson portunus* and *Enterospora nucleophila*, have been identified to infect farmed crabs and fish, respectively [9,10]. Among immunocompromised individuals, *Enterocytozoon bieneusi* and *Encephalitozoon intestinalis* are the most common microsporidia infections in humans and are associated with diarrhea and systemic illness [11,12]. In underdeveloped countries, up to 51% of HIV-infected individuals with diarrhea have microsporidia infection [13]. Infections with zoonotic microsporidia have been discovered in livestock, companion animals, and wildlife, which poses a risk to public health [14]. In recent decades, the NIAID and CDC have recognized that many of these species of microsporidia pose a threat to human health. Thus, microsporidia have been classified as Category B priority pathogens for biodefense research [15].

There are few effective therapeutic interventions available to treat microsporidia infections [16]. Albendazole is known to inhibit microtubule polymerization by binding beta-tubulin and is effective against a variety of parasites with little adverse effects in humans [17,18]. In infections of *Encephalitozoon* spp., albendazole has been demonstrated to control microsporidiosis [19]. However, several studies have shown that albendazole has a limited effect on microsporidiosis caused by *E. bieneusi* and *Vittaforma cornea* and that these species encode beta-tubulin with substitutions associated with resistance [20–22]. Analogs of albendazole, benomyl, and carbendazim have been used to control microsporidia in insects such as *Nosema heliothidis* in *Heliothis zea* and *Nosema kingi* in *Drosophila willistoni* [16,23,24]. However, microsporidiosis in mammals cannot be treated with benomyl and carbendazim because of their hepatotoxicity and toxic effects on reproduction [25–27]. Since benzimidazoles have been widely used for decades, resistance fears have arisen. Fumagillin binds specifically and covalently to methionine aminopeptidase type 2 (MetAP2) and can inhibit many species of microsporidia [28,29]. However, due to concerns with host toxicity, this drug is not approved for use in humans [30]. Furthermore, fumagillin has been associated with concerns regarding its toxicity in agriculture applications [31]. Therefore, there is a need to identify additional microsporidia inhibiting agents.

The model organism *Caenorhabditis elegans* has become a useful system in which to study microsporidia infections and to identify inhibitors. The first microsporidia reported to infect *C. elegans* was *Nematocida parisii* [32]. Infection of *C. elegans* by *N. parisii* begins when spores are ingested into the worm's intestinal lumen, where they expel their unique invasion apparatus called the polar tube [33]. This causes the sporoplasm to be deposited inside intestinal cells. The parasite then proliferates intracellularly as meronts and then differentiates into spores,

which then exit into the intestinal lumen. Infection of *C. elegans* with *N. parisii* results in smaller body size, reduced reproductive fitness, and shortened life span [34–36]. This host-parasite system has become a model in which to study mechanisms of microsporidia invasion, proliferation, and spore exit [35,37–39]. Other species of microsporidia have also been found to infect *C. elegans* including *Pancytospora epiphaga*, which is related to the human-infecting species *E. bieneusi* and *V. cornea* [40–42]. We recently described the use of *C. elegans* to screen compounds for activity towards *N. parisii* using a high-throughput 96-well based assay, resulting in the identification of inhibitors of microsporidia invasion and proliferation [4,37,43].

The Pandemic Response Box (PRB) is a promising source of anti-microsporidia compounds. Developed by the Medicines for Malaria Venture with the support of the Drugs for Neglected Disease Initiative, this is an open-access compound library consisting of 400 small molecule compounds that have antifungal, antibacterial and antiviral activity [44]. Compounds from this collection include those that can inhibit other types of parasites such as nematodes, amoebas, and the causative agent of malaria [45–47].

To identify novel microsporidia inhibitors, we screened the PRB using a modified version of our previously described *C. elegans-N. parisii* assay [43]. We screened this compound collection at two concentrations and identified four chemical inhibitors of microsporidia infection. We validated these compounds and additionally showed that all four compounds reduced the pathogen load of *N. parisii*. We then studied at what stage in the microsporidia life cycle each compound is active. We found that MMV1782387, an analog of albendazole, inhibits microsporidian proliferation. This compound, albendazole, and the other MMV compounds we identified also inhibit *P. epiphaga*. We also show that MMV1006203 inhibits spore firing and related flavone analogs also prevent *N. parisii* infection. Additionally, the viability of spores is decreased when they are treated with MMV1593539. Together our study identifies additional microsporidia inhibitors that can block microsporidia proliferation and invasion.

## Material and methods

### C. elegans maintenance

The food source for *C. elegans*, *Escherichia coli* OP50-1, was grown to saturation in lysogeny broth (LB) for 18 hours at 37˚C. To generate animals for infection assays, the wild-type *C. elegans* strain N2 was grown as a mixed population and L4 stage worms were picked onto 10 cm nematode growth media (NGM) plates seeded with 10x OP50-1 *E. coli* and kept at 21˚C for 4 days [48]. In order to synchronize the worms, M9 solution was used to remove the worms from the NGM plates, sodium hypochlorite and sodium hydroxide were used to bleach them. Embryos from gravid adults were released into the solution and after washing were incubated at 21˚C for 18 to 24 hours until the embryos hatched.

### N. parisii spore preparation

*N. parisii* (ERTm1) spores were prepared as described previously [36]. *C. elegans* N2 worms were infected with *N. parisii* spores on NGM plates. Worms were incubated for several days to generate a large population of infected worms, which were harvested and frozen at -80˚C. Zirconia beads (2 mm diameter) were used for mechanical disruption of the infected worms, followed by the removal of embryos, larvae, and debris using a 5 μm filter (Millipore). *N. parisii* spore preparations were confirmed to be free of contaminated bacteria and stored at -80˚C. Spore concentration was measured by counting DY96-stained spores using a sperm counting slide (Cell-VU).

## Source of chemicals

Medicines for Malaria Venture (MMV, Geneva, Switzerland) provided the PRB, which contains compounds dissolved in 10 μL of DMSO at a concentration of 10 mM for inclusion. 2 μL of the stock compounds were transferred to new plates containing 3 μL DMSO to generate 4 mM stocks. For retesting, the individual solid compounds of MMV1006203, MMV1593539, MMV1634497 and MMV1782387 were provided by the MMV. The analog compounds of albendazole, thiabendazole, carbendazim, oxfendazole, mebendazole, fenbendazole, flavone and displurigen were purchased from MilliporeSigma. Stocks of all compounds were stored at -80˚C.

## Phenotypic assays in 96-well plates to identify microsporidia inhibitors

Previously described methods were adapted to quantify the ability of compounds to restore the ability of *C. elegans* to produce progeny in the presence of *N. parisii* [43]. Each well of a 96-well plate was filled with 25 μL of K-medium (51 mM NaCl, 32 mM KCl, 3 mM $CaCl_2$, 3 mM $MgSO_4$, 3.25 μM cholesterol) containing 5x OP-50 and *N. parisii* spores, with the exception of column 12 to which spores were not added. 25 μL of K-medium containing L1 worms was then added to each well. 500 nL of compounds from the PRB were pinned into columns 2–11 using a 96-well pinning tool manufactured by V&P Scientific. Additionally, 500 nL of DMSO was added to columns 1 and 12 for infected and uninfected controls, respectively. Each well contained 100 bleach-synchronized L1 worms, 1% DMSO, 15,000 *N. parisii* spores/μL and 100 μM or 40 μM of compounds. A breathable adhesive porous film was used to cover the 96-well plates, which were placed inside humidity boxes wrapped in parafilm and incubated for 6 days at 21˚C with shaking at 180 rpm. Each compound was tested three times at both concentrations, with the exception of 17 compounds for which there was not sufficient amounts at which to test at 100 μM and 9 compounds for which there was not sufficient amounts at which to test at 40 μM (See S1 Data).

## Quantification of progeny production

Following incubation, 10 μL of 0.3125 mg/mL Rose Bengal solution was added to each well using an Integra VIAFLO 96 Electronic pipette. Plates were then wrapped in parafilm and incubated for 16–24 hours at 37˚C, resulting in magenta staining of the worms. 240 μL M9/0.1%Tween-20 was added to each well and the plate and centrifuged for 1 minute at 2200 x g. 200 μL supernatant was removed from each well and 150 μL of M9/0.1%Tween-20 was added to each well. Upon mixing up the worms in the plate, 25 μL of the worms were transferred to 96-well white clear bottom plates containing 300 μL M9/0.1%Tween-20. After 30 minutes, plates were scanned using an Epson Perfection V850 Pro flat-bed scanner with the following settings: positive film-holder, 4800 dpi, and 24-bit color. In order to highlight stained worms, images were also modified using GIMP version 2.8.18, with horizontal and vertical gridlines positioned such that each well is separated by a grid and removing HTML color codes #000000 and #FFC9AF. Images were also modified by applying unsharp masking with the following parameters (radius = 10, effect = 10, threshold = 10). Hue saturation was adjusted by changing the lightness to 100 and the saturation to -100 for yellow, blue, cyan, and green. For red and magenta, the lightness was changed to -100 and the saturation to 100. Each well was exported as a single.tiff image using LZW compression. MATLAB was used to run WorMachine [49] with the pixel binarization threshold set to 30, the neighboring threshold to set to 1, and the maximum object area set toto 0.003%.

### Continuous infection assays

24-well assay plate containing a total volume of 400 μL including 800 L1 worms and 15,000 *N. parisii* spores/μL were used for continuous infection assays. Assays were performed for three biological replicates using 100 μM of each compound except for dexrazoxane (60 μM). During the incubation period, test plates were covered with breathable adhesive porous film, the boxes were enclosed in parafilm, and the plates were incubated at 21˚C, with shaking at 180 rpm for four days. Incubated samples were washed with M9/0.1%Tween-20, acetone-fixed, DY96-stained, and analyzed by fluorescence microscopy.

### Pulse infection assays

To generate infected worms, ~8000 bleach-synchronized L1 worms, 30 million *N. parisii* spores, and 5 μL 10x OP50-1 were added to 6 cm NGM plates and incubated for three hours at 21˚C after drying. To remove excess spores, the worms were washed twice with 5 mL M9/0.1% Tween-20. Worms were then added to 24-well plates and set up as described in the continuous infection assays, with the exception that no spores were added. For each of the biological replicates, three wells were assayed for each compound. After incubation for 2 or 4 days as described above, samples were fixed in acetone and stained with DY96 and a FISH probe as described below.

### Spore firing assays

Spores at a concentration of 30,000 spores/μL were incubated for 24 h at 21˚C with compounds at a concentration of 200 μM, except for ZPCK was at 120 μM, and 2% DMSO. After being washed three times with 1 mL K-medium, the spores were used in the 24-well assay plates as described above. The final concentrations in these assays were 15,000 spores/ μL, 100 μM compounds except ZPCK (60 μM), and 1% DMSO. Each compound was tested in three biological replicates in all assays. Incubation was performed as described above and after 3 h, samples were fixed in acetone, stained with FISH and DY96, and examined by fluorescence microscopy.

### Mortality assay

*N. parisii* spores were incubated for 24 h at 21˚C with compounds at a concentration of 200 μM. For the heat treatment control, spores were incubated at 100˚C for 10 minutes. The spores were washed twice with $H_2O$, resuspended in 100 μL of $H_2O$ containing 2 mg/L Calcofluor White M2R and 8 μM SYTOX Green nucleic acid stain, and incubated for 10 minutes at room temperature. Spores were washed twice in $H_2O$, 2.5 μL of each mixture were spotted on slides containing 2% agar. The mortality rate was determined by counting the percentage of calcofluor white stained spores that contained SYTOX Green signal.

### DY96 staining, fluorescence in situ hybridization (FISH), and fluorescence microscopy

To remove excess OP50, samples were washed twice in 1 mL M9/0.1%Tween-20. They were fixed in 700 μL acetone for 15 minutes or 500 μL PFA solution (4% PFA, 1x PBS, 0.1% Tween-20) for 30 minutes. Then, samples were washed twice in PBS/0.1%Tween-20. For DY96 staining, 500 μL DY96 staining solution (10 μg/μL DY96, 0.1% SDS in 1xPBS + 0.1% Tween-20) was added and samples were rotated for 30 minutes. EverBrite Mounting Medium with DAPI was then added to the samples. FISH was performed using the microB FISH probe for *N. parisii* 18S rRNA (ctctcggcactccttcctg) conjugated to Cal Fluor Red 610 (LGC Biosearch

Technologies) [32]. After washing with PBS/0.1%Tween-20, samples were incubated in hybridization buffer (900 mM NaCl, 20 mM pH = 8 Tris HCl, 0.01% SDS) containing 5 ng/μL FISH probe at 46˚C for 1–6 hours. Samples were washed once with 1 mL wash buffer (50 mL hybridization buffer + 5 mM EDTA). These samples were also stained with DY96 as described above, except with DY96 at 20 μg/μL. Samples were imaged using a ZEISS Axio Imager 2 at 5x–63x magnification and images were captured using Zeiss Zen 2.3. Gravid worms were defined as the proportion of animals containing any number of embryos. Infected worms were defined as the proportion of animals displaying any newly formed spores. Low infection was defined as spores present in less than one half of an animal, moderate infection was defined as spore present in half of an animal, and high infection was defined as spores present throughout both halves of an animal. L1 progeny of the parents could be distinguished in size and were not included in these measurements.

### P. epiphaga infection assays

*P. epiphaga* strain JUm1396 spores were prepared similar to as described above for *N. parisii*. For infection experiments with *P. epiphaga*, 24-well assay plates were set up to contain a final volume of 400 μL in K-medium, including 800 L1 worms and 80,000 *P. epiphaga* spores /μL. All compounds were at a concentration of 100 μM, except for dexrazoxane which was used at 60 μM. The final concentration of DMSO was 1%. Plates were incubated as described above for four days. The samples were fixed in 4% PFA and stained by the FISH probe specific to *P. epiphaga* 18S rRNA (CAL Fluor Red 610CTCTATACTGTGCGCACGG). Fluorescence microscopy and quantification of FISH fluorescence were performed as described above.

### Statistical analyses

The data were collected from three independent experiments with 3 biological repelicates and analyzed by GraphPad Prism. The means between replicates were compared using either Student's t-test or one-way ANOVA with post hoc correction.

## Results

### Screen of Pandemic Response Box identifies 4 microsporidia inhibitors

To identify compounds from the PRB which inhibit microsporidia infection, we adapted our previously described 96-well infection assays [43]. Earliest larval (L1) stage of *C. elegans* were incubated with *N. parisii* spores and compound in liquid for six days at 21˚C. Host animals infected with *N. parisii* produce a reduced number of offspring and compounds which inhibit microsporidia can restore the ability of these animals to produce progeny. To quantify the number of offspring produced when incubated with each compound, animals in each well were stained with rose bengal, imaged with a flatbed scanner, and counted using WorMachine [49] (see methods). Each compound was screened in triplicate at a concentration of 100 μM. We identified six compounds that resulted in progeny production in infected worms of 35–90% relative to the uninfected controls (Fig 1A). These compounds also increased progeny production by 1.7–8.5-fold compared to infected controls. Four of these compounds significantly improved the production of *C. elegans* progeny in the presence of *N. parisii* (Fig 1B–1D). To determine the effect of the compound collection at a lower concentration, we screened each compound at 40 μM. We observed three compounds which increased progeny production over 3–5.3-fold more than infected controls and to more than 13% of the uninfected controls (Fig 1E). Two of these compounds resulted in significantly more progeny production at 40 μM and these two compounds were also observed to be significant at 100 μM (Fig 1F and

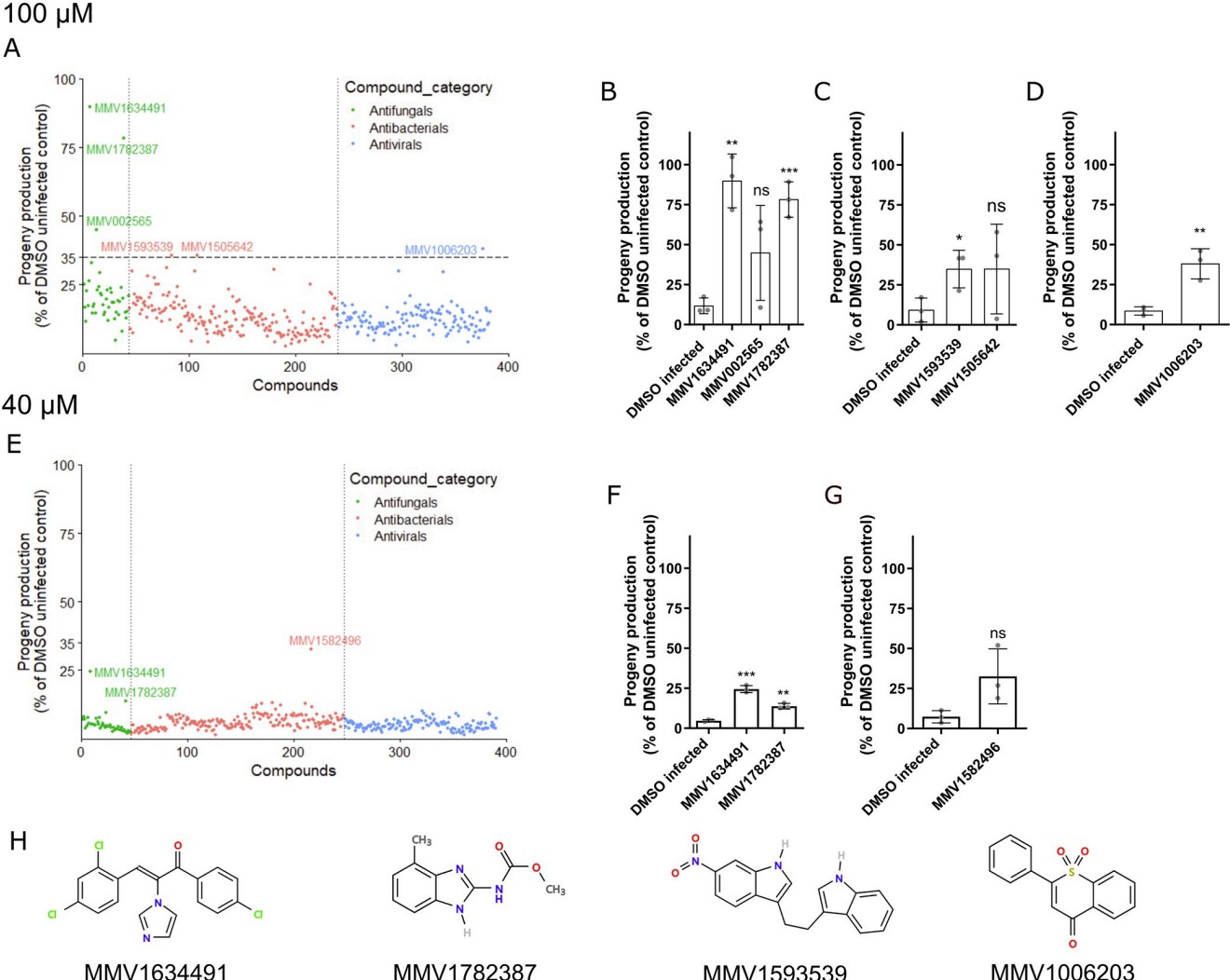

**Fig 1. Four compounds from the PRB restored *C. elegans* progeny production in the presence of *N. parisii*.** (A and E) Compounds at concentration of 100 μM (A) or 40 μM (E) were incubated with *C. elegans* and *N. parisii* for 6 days. Each point represents the mean progeny production of a compound expressed as the percentage of the DMSO uninfected control. The compound-ID is shown for compounds that had an activity of at least 35% (A) or ≥ 13% (E). Compounds are divided into their disease area as classified by the PRB and colored according to the legend at the right. 17 compounds in the collection were not screened at 100 μM and 9 compounds not screened at 40 μM due to lack of material (S1 Data). (B-D) Compounds that had an activity of at least 35% at 100 μM. (B) Antifungals, (C) Antibacterials, (D) Antivirals. (F and G) Compounds that had an activity ≥ 13% at 40 μM. (F) Antifungals, (G) Antibacterials. (H) The chemical structures of the four PRB compounds with significant activity. Statistical significance was determined by Student's t-test with comparisons to the DMSO infected control. Means ± SD (horizontal bars) are shown. (*p < 0.05, **p < 0.01, ***p < 0.001, ns means not significant).

1G). In total we identified four compounds with significant activity which we validated and characterized in subsequent experiments (Fig 1H).

We next sought to determine whether the four compounds which significantly restored *C. elegans* progeny production also limited *N. parisii* infection. We set up assays similar to our initial screen by culturing L1 worms continuously with *N. parisii* spores in the presence of compounds in 24-well plates. After four days, worms were fixed and then stained with direct yellow 96 (DY96), which binds to chitin, a critical component of *N. parisii* spore walls and *C. elegans* embryos (Fig 2A) [42,50,51]. In the presence of *N. parisii* spores, all four compounds significantly increased the proportion of adult worms containing embryos (Fig 2B), which is

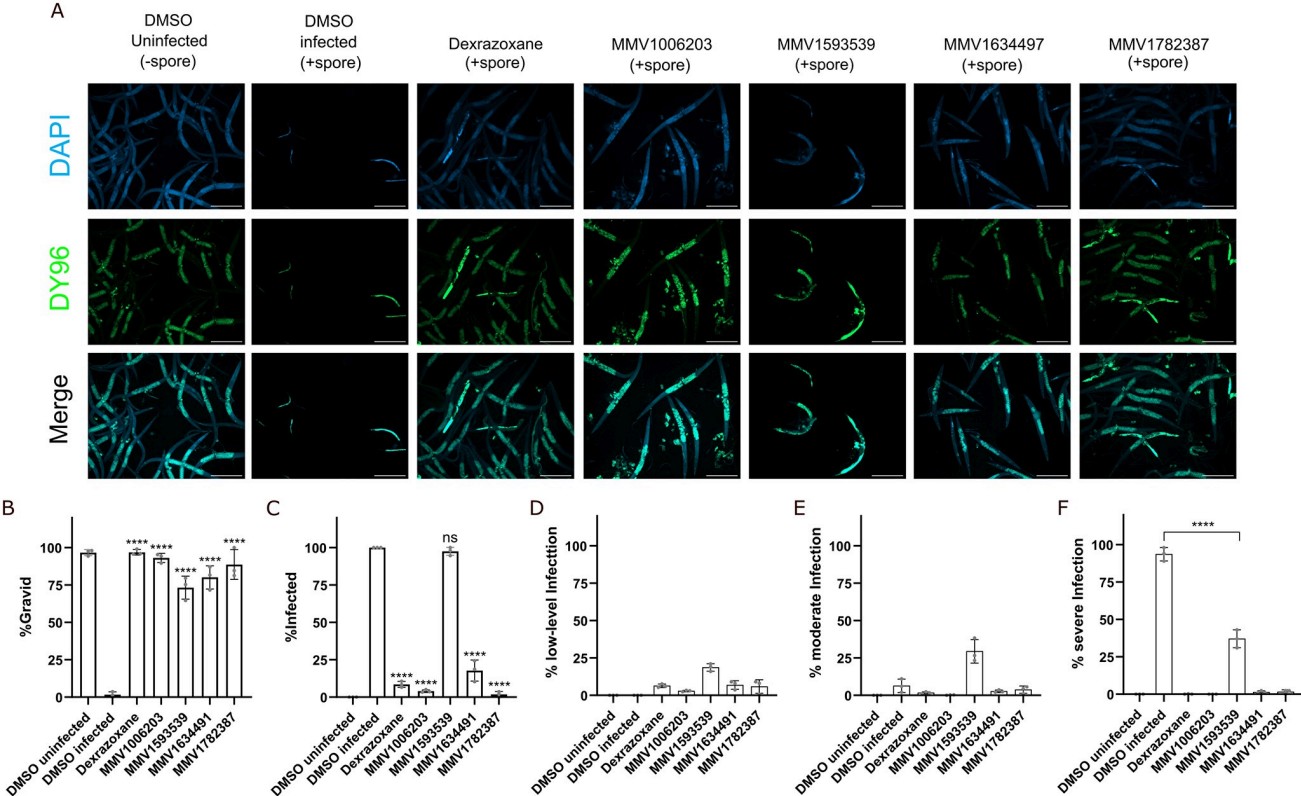

**Fig 2. Identified MMV compounds inhibit *N. parisii*.** (A-F) L1 stage animals with incubated in the presence of *N. parisii* spores and compounds for 4 days. Animals were then fixed and stained with DY96 and DAPI. (A) Representative images of continuous infection assays taken at a magnification of 50x; scale bars are 500 μm. (First and second columns) Worms incubated without (-spore) or with spores (+spore). As a result of microsporidia infection, fewer worms become gravid, and new spores are formed. Microsporidia spores and nematode embryos are stained green by DY96. (Third to seventh columns) Dexrazoxane and the identified PRB compounds inhibit spore formation and restore embryo production. (B) The percentage of worms with embryos (n = 3, N = ≥ 100 worms counted per biological replicate). (C) The percentage of worms with newly formed spores (n = 3, N = ≥ 100 worms counted per biological replicate). (D-F) The percentage of worms with (D) low-level infection, (E) moderate infection, and (F) severe infection. The P-values were determined by one-way ANOVA with post hoc test. Means ± SD (horizontal bars) are shown. (***p < 0.001, ****p < 0.0001, ns means not significant).

consistent with the results from our initial screen. Treatment with the newly discovered compounds and the known microsporidia inhibitor dexrazoxane significantly reduced animals displaying newly formed *N. parisii* spores, except for MMV1593539 (Fig 2C) [43]. To determine if MMV1593539 had any impact on *N. parisii* infection, we quantified the proportion of worms displaying a low, moderate, or high infection level (See methods) (Fig 2D–2F). Under these infection conditions, ~90 percent of control worms displayed high infection levels and all four compounds significantly lowered infection (Fig 2F).

## MMV1782387 inhibits the proliferation of *N. parisii*

Inhibition of microsporidia infection could be achieved through two mechanisms. One, by preventing microsporidia from invading cells, which could occur either through the inactivation of spores or by preventing spores from germinating. Two, acting after invasion to reduce proliferation. To test whether compounds limited proliferation, we set up pulse-chase experiments where we infected worms for three hours, washed away excess spores, and then incubated the worms with one of the four MMV compounds or with dexrazoxane which was previously shown to limit *N. parisii* proliferation (Fig 3A) [43]. Treatment with MMV1782387

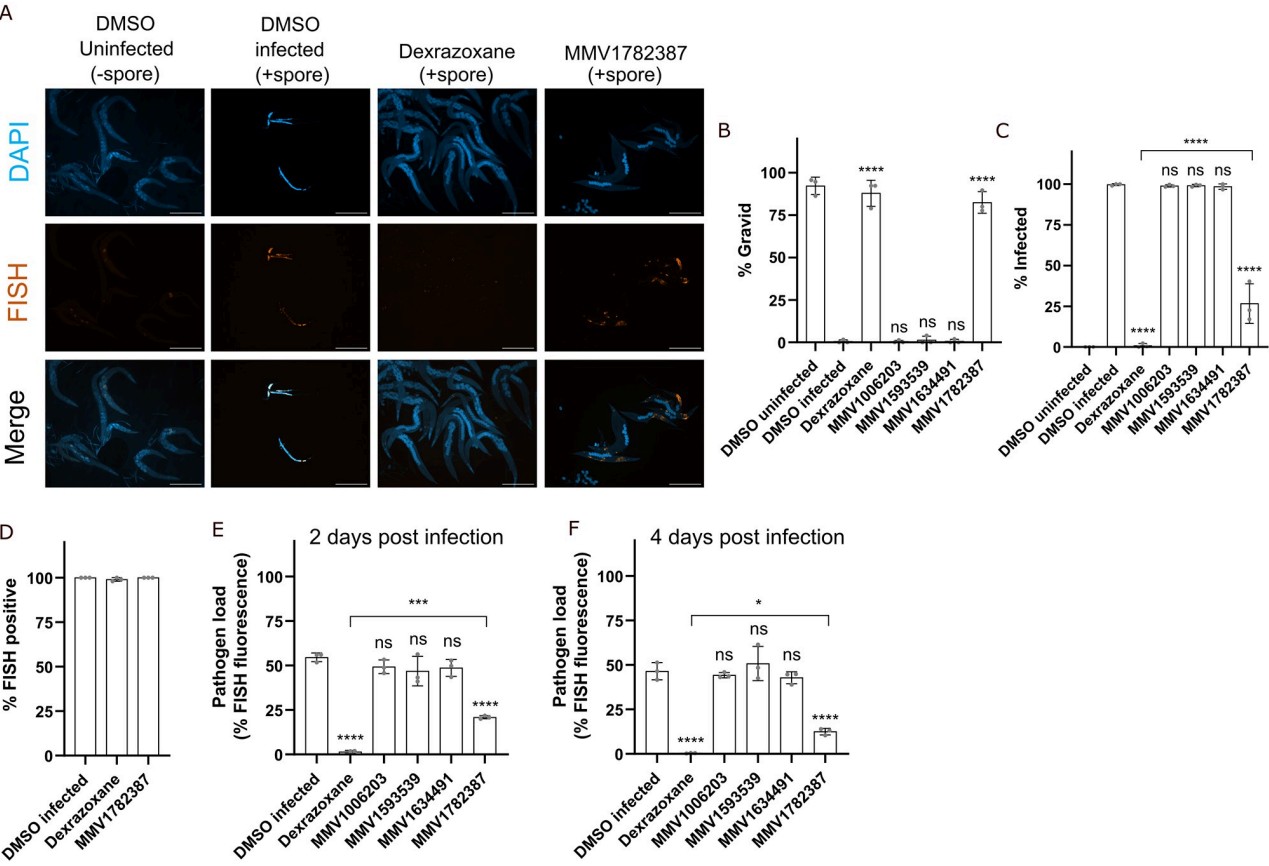

**Fig 3. MMV1782387 inhibits microsporidia proliferation.** (A-F) L1 stage animals were incubated in the presence of *N. parisii* spores for 3 hours and then washed to remove excess spores. Compounds were then added, and animals were incubated for 2 (E) or 4 (A-D and F) days, fixed, and stained with DY96, DAPI and a FISH probe specific to the *N. parisii* 18S rRNA. (A) Representative images of pulse infection assays taken at a magnification of 50x; scale bars are 500 µm. (First and second columns) Worms uninfected or infected with *N. parisii* spores. Sporoplasms and meronts are stained in red with FISH probes. (Third and fourth columns) Dexrazoxane or MMV1782387 treatment reduces *N. parisii* meronts. (B) The percentage of worms with embryos (n = 3, N = ≥ 100 worms counted per biological replicate). (C) The percentage of worms with newly formed spores (n = 3, N = ≥ 100 worms counted per biological replicate). (D) The percentage of worms with FISH signal (n = 3, N = ≥ 100 worms counted per biological replicate). (E and F) Quantitation of pathogen load (FISH fluorescence area %) per worm for (E) 2 or (F) 4 days post infection (n = 3, N = 10 animals quantified per biological replicate). The P-values were determined by one-way ANOVA with post hoc test. Means ± SD (horizontal bars) are shown. ($^{*}$p < 0.05, $^{***}$p < 0.001, $^{****}$p < 0.0001, ns means not significant).

and dexrazoxane increased the gravidity of worms (Fig 3B). Treatment with MMV1782387 also inhibited *N. parisii* proliferation with 26.7% of worms having newly formed spores, which is less inhibition than was observed with dexrazoxane (0.89%) (Fig 3C).

To determine whether MMV1782387 inhibits *N. parisii* by slowing proliferation or enhancing parasite clearance, we examined pulse-chase infected animals at either 2 days (before spore formation) or 4 days (after spore formation) post infection with probes specific for *N. parisii* 18S rRNA [32]. Dexrazoxane was previously shown to significantly reduce the pathogen burden within animals, without influencing the proportion of infected animals [43]. We observed that MMV1782387, similar to dexrazoxane, did not cause a reduction in infected animals, but significantly reduced the amount of meronts in the worms (Fig 3D–3F). None of the other MMV compounds we tested reduced pathogen load of the worms at either 2- or 4-days post treatment. Dexrazoxane also caused significantly more reduction in pathogen levels compared

to MMV1782387 (Fig 3E and 3F). Together our results show MMV1782387 inhibits microsporidia proliferation but does not enhance parasite clearance.

## *N. parisii* spore firing is inhibited by MMV1006203

The initial step of microsporidia invasion is spore germination. Many microsporidia, including *N. parisii*, germinate (also called spore firing) in the intestinal lumen to initiate infection [32,52,53]. In order to determine whether any compounds reduce germination, we conducted spore firing assays using the four MMV compounds as well as ZPCK, which has previously been shown to inhibit spore germination [43]. *N. parisii* spores were incubated with compounds for 24 hours and the spores were then washed to remove the compounds. These spores were then cultured with *C. elegans* at the L1 stage for three hours and stained with FISH to visualize the sporoplasms and DY96 to visualize the spores. Spore firing was determined by counting the number of spores that did not contain a sporoplasm divided by the total number of spores. ZPCK and one of the MMV compounds, MMV1006203, significantly reduced the spore firing rate (Fig 4A). All of the compounds, as well as ZPCK, also significantly reduced the number of sporoplasms that had invaded each worm (Fig 4B).

## *N. parisii* spores are inactivated by MMV1593539

One way that microsporidia infection could be prevented is through the inactivation of microsporidia spores. To test whether any of the MMV compounds inactivated the spores, we performed mortality assays. Compounds were incubated with *N. parisii* spores for 24 hours and then stained with calcofluor white, a dye that binds to spore wall, and SYTOX Green, a dye that stains the nucleus of inviable cells. We then counted the number of spores that were inviable, using heat treatment as a control for maximum spore inactivation. One of the MMV compounds, MMV1593539, significantly increased, mortality rates, though not to the same extent as heat treatment (Fig 4C).

## Benzimidazole and flavone analogs inhibit *N. parisii* infection

The benzimidazole molecule albendazole is one of the main treatment options currently used for microsporidia infection. The structure of MMV1782387 is similar to albendazole and the

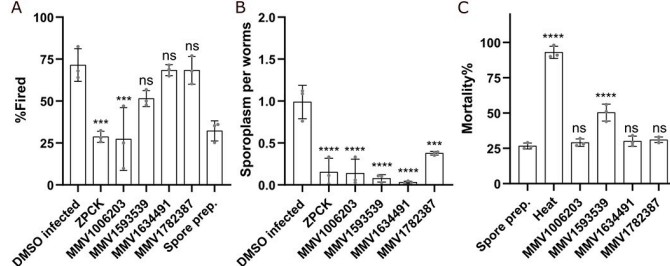

**Fig 4. MMV1006203 inhibits spore firing *in vivo* and MMV1593539 inactivates spores *in vitro*.** (A-B) *N. parisii* spores were incubated with compounds for 24 hours and then washed to remove compounds. Spores were then incubated with L1 stage worms for 3 hours, fixed, and stained with DY96 and *N. parisii* 18S rRNA FISH probe. (A) The percentage of fired spores in the intestinal lumen (n = 3, N = ≥ 50 spores counted per biological replicate). (B) The mean number of sporoplasms per worm (n = 3, N = ≥ 50 worms counted per biological replicate). (C) *N. parisii* spores were incubated with compounds for 24 hours and stained with SYTOX Green and Calcofluor White M2R. The percentage of spores that showed SYTOX Green staining (n = 3, N = ≥ 100 spores counted per biological replicate). The P-values were determined by one-way ANOVA with post hoc test. Means ± SD (horizontal bars) are shown. (***p < 0.001, ****p < 0.0001, ns means not significant).

PRB contains several other benzimidazoles including carbendazim, fenbendazole, and oxfendazole. However, these other benzimidazole compounds were not identified in our initial screen. We first tested whether MMV1782387 and six benzimidazole analogs reduced the reproductive fitness of uninfected *C. elegans* (Fig 5G). At 40 μM, there is no difference in the percentage of worms forming embryos when treated with any of the compounds (Fig 5A). In contrast, menbendazole and oxfenfazole significantly reduced progeny production in uninfected animals at 100 μM, indicating moderate toxicity to the host (Fig 5D). We then tested whether these benzimidazole compounds could restore the formation of embryos in animals infected with *N. parisii* using our continuous infection assays. Except thiabendazole, all of the compounds increased the percentage of gravid worms in the presence of *N. parisii* at 40 μM (Fig 5B). At 100 μM, MMV1782387, albendazole, and carbendazim increased the percentage of gravid worms with *N. parisii* (Fig 5E). MMV1782387-treated worms displayed the largest increase in the percentage of gravid worms. We also examined the percentage of worms with newly formed spores when treated with these compounds. Except thiabendazole, all compounds displayed a reduction in infected animals at 40 μM (Fig 5C). All compounds displayed a reduction in infected animals at 100 μM, with MMV1782387 displaying amongst the strongest inhibition of infection (Fig 5F). These results suggest that benzimidazoles can inhibit *N. parisii* and that MMV1782387 shows both strong inhibition of *N. parisii* as well as low host toxicity in *C. elegans*.

The compound we identified which inhibits spore firing, MMV1006203, has a structure similar to flavone. To test whether flavone and an analog of MMV1006203, displurigen, could inhibit *N. parisii*, we tested these compounds in continuous infection assays at a concentration of 40 μM or 100 μM (Fig 5L). All three of these compounds significantly increased the proportion of gravid worms and reduced infection rates at a concentration of 40 μM (Fig 5H and 5I). At a concentration of 100 μM, all three molecules were effective at inhibiting microsporidia infection, though only MMV1006203 could significantly increase the gravidity of worms (Fig 5J and 5K). These results show that molecules based on a flavone structure can inhibit *N. parisii* infection.

## Identified MMV compounds inhibit *P. epiphaga*

To test whether the four MMV compounds we identified were effective against other microsporidia species, we tested them against *P. epiphaga*. This species of microsporidia infects the hypodermis and muscle of *C. elegans* and belongs to the *Enterocytozoonida* clade, along with the human pathogens *V. cornea* and *E. bieneusi* [40,42]. In order to examine whether the four inhibitors we identified from the PRB could inhibit *P. epiphaga* infection of *C. elegans*, we used FISH staining to quantify the pathogen load. All of the compounds significantly reduced *P. epiphaga* infection levels (Fig 6A). We tested albendazole against *P. epiphaga* and observed that this compound also inhibited infection (Fig 6B).

## Discussion

We screened the open-access PRB compound library, identifying four compounds with antimicrosporidia activity. We quantified the ability of compounds to reproducibly alleviate the reduction in *C. elegans* progeny caused by *N. parisii* infection. We validated all four compounds identified from the initial screen, which is an improvement over our previous screen where we qualitatively determined the effect of compounds from a single replicate and only about half of the initially identified compounds were validated [43]. We show that the compounds we identified have different effects on microsporidia, with MMV1782387 preventing proliferation, MMV1006203 preventing spore firing, and MMV1593539 causing an increase

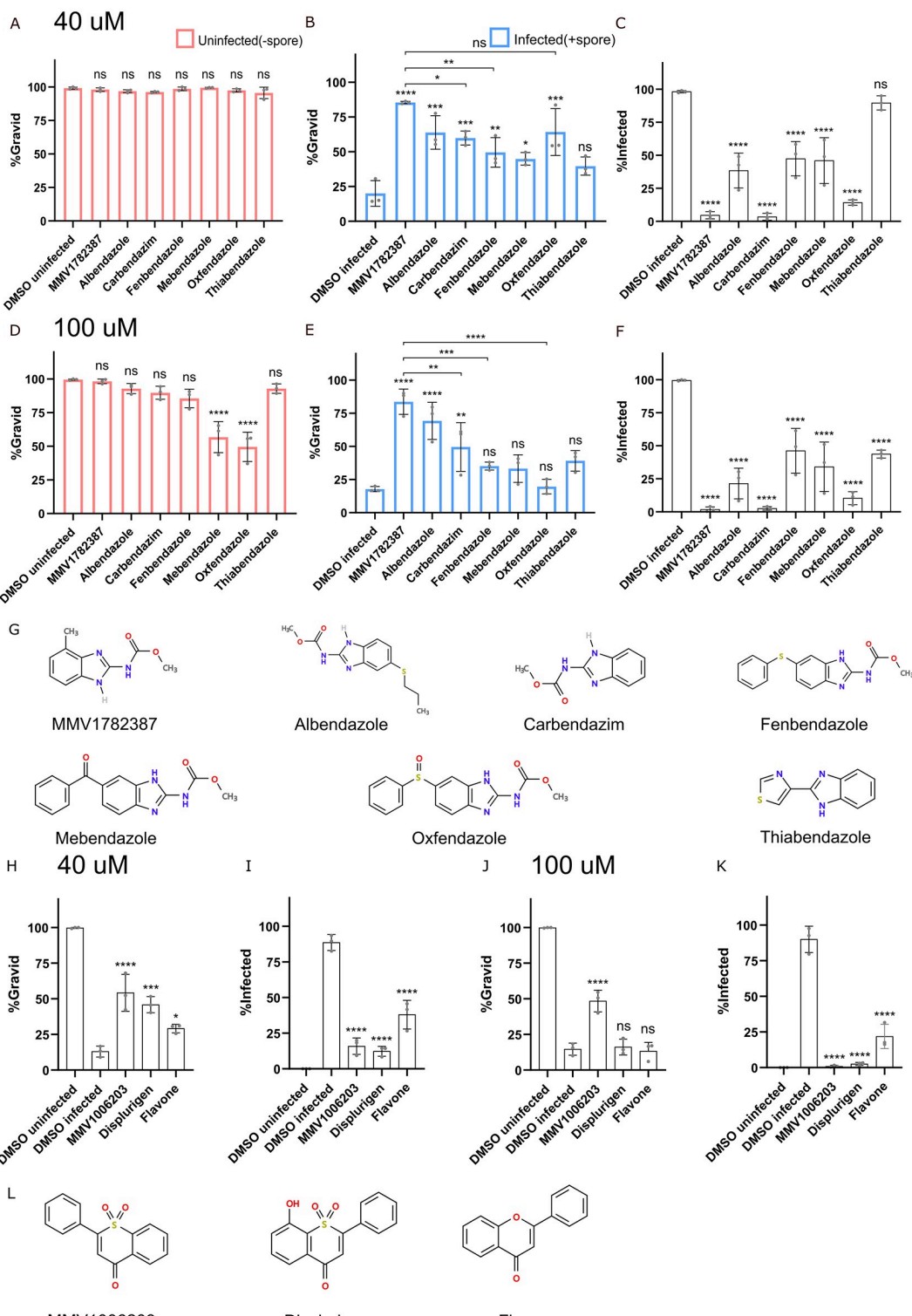

**Fig 5. Benzimidazole and flavone analogs limit *N. parisii* infection.** (A-F and H-K) L1 stage animals were continuously incubated with *N. parisii* spores and indicated compounds for 4 days, fixed, and stained with DY96 and DAPI. (A-F) Effect of benzimidazoles at 40 μM (A-C) or 100 μM (D-F) on the percentage of worms with embryos in uninfected worms (A and D), infected worms (B and E) and the percentage of worms (C and F) with newly formed spores (n = 3, N = ≥ 100 worms counted per biological replicate). (G) Chemical structures of benzimidazole analogs. (H-K) Effect of flavones at 40 μM (H-I) or 100 μM (J-K) on the percentage of worms with embryos (H and J) and the percentage of worms with newly formed spores

(I and K) (n = 3, N = $\geq$ 100 worms counted per biological replicate). (L) Chemical structures of flavone analogs. The P-values were determined by one-way ANOVA with post hoc test. Means ± SD (horizontal bars) are shown. (*p < 0.05, **p < 0.01, ***p < 0.001, ****p < 0.0001, ns means not significant).

in spore mortality. When used to treat spores, all four compounds inhibit invasion, though it is not clear why MMV1634491 and MMV1782387 reduce sporoplasm numbers. It appears that these compounds limit microsporidia invasion, but through some mechanism that will require additional experiments to determine. All four compounds we identified in this study limited both *N. parisii* and *P. epiphaga* infection. These results further demonstrate that *C. elegans* can be used to efficiently identify compounds with activity against multiple species of microsporidia. One limitation of our approach is that inhibitors that reduce the reproductive fitness of *C. elegans* on their own will not be observed. However, this is also potentially beneficial as host toxicity, at least within the *C. elegans* context, is evaluated at the same time as inhibition of microsporidia infection.

There is a range of bioactivity associated with benzimidazoles, including anti-inflammatory, antihypertensive, anti-bacterial, anti-parasitic, and anti-fungal properties [16,54–56]. The benzimidazole albendazole is one of the most common treatments for microsporidia. Here we show that MMV1782387, a benzimidazole carbamate, has amongst the strongest inhibition of *N. parisii* and relatively low host toxicity in *C. elegans*. Benzimidazoles are known to inhibit *C.*

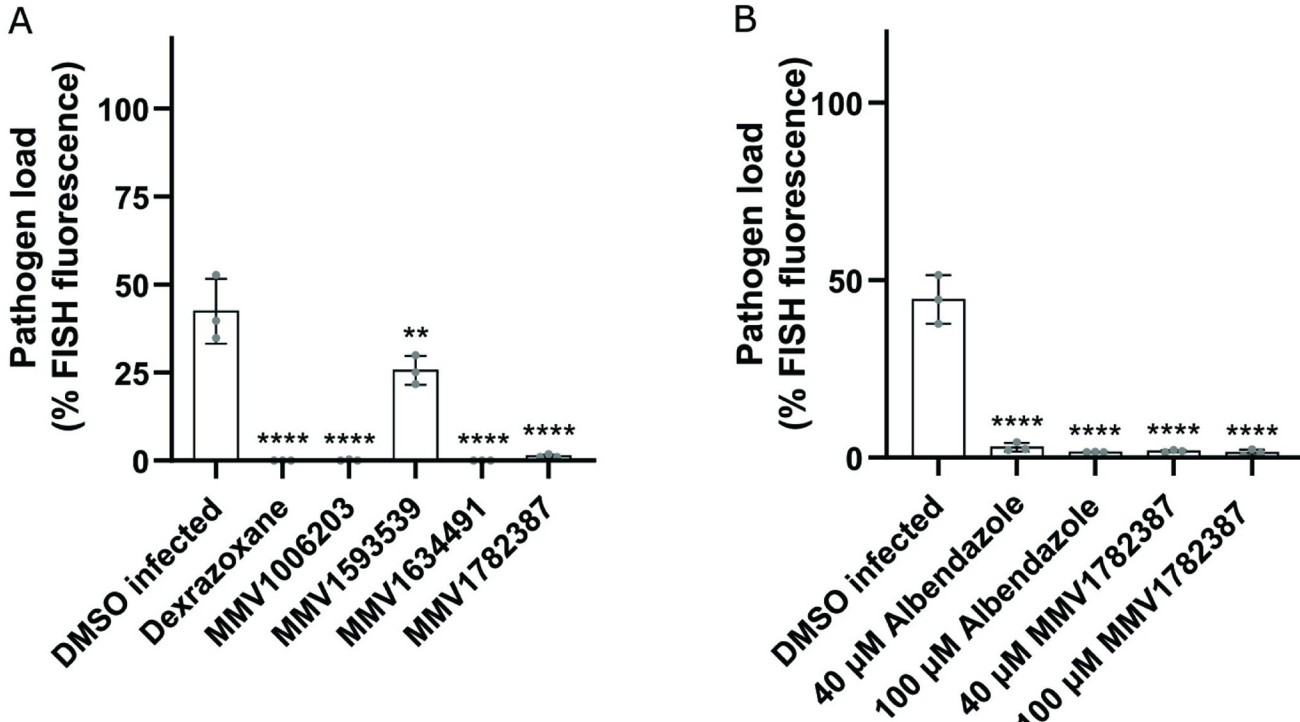

**Fig 6. *P. epiphaga* infection is impeded by the identified MMV inhibitors and albendazole.** (A and B) L1 stage animals were continuously incubated with *P. epiphaga* spores for four days, fixed, and stained with a *P. epiphaga* 18S rRNA FISH probe. Quantification of pathogen load (%FISH fluorescence area) per worm 4 days post infection (n = 3, N = 10 animals quantified per biological replicate). The P-values were determined by one-way ANOVA with post hoc test. Means ± SD (horizontal bars) are shown. (*p < 0.05, **p < 0.01, ***p < 0.001).

*elegans* and natural resistance to these compounds has arisen through genetic variation in beta-tubulin [57,58]. Host toxicity has been shown to be mediated through inhibition of neuronal beta-tubulin [59]. Carbendazim, fenbendazole and oxfendazole are present in the PRB, however, in our preliminary screening, we did not find that these compounds improved progeny production in the presence of *N. parisii*. Beta-tubulin is the likely target of albendazole in microsporidia. *V. cornea* and *E. bieneusi* contain a glutamine at position 198 in beta-tubulin that is associated with albendazole resistance, and mutations in this position provide resistance in *C. elegans* [60]. *P. epiphaga* beta-tubulin encodes for glutamate at this position which is associated with albendazole sensitivity and is consistent with our data showing that this species can be inhibited by benzimidazoles [21,22,41]. Given the similarity of the compounds, MMV1782387 may also inhibit beta-tubulin. In compound library screens, multistage activity is one of the most preferred attributes for molecules which inhibit pathogens [61]. We show that MMV1782387, which inhibits proliferation of *N. parisii* can also reduce invasion by about 50%. MMV1782387 has been shown to be effective against several fungal pathogens which cause eumycetoma and may have potential for further development as an inhibitor of fungal pathogens [62].

Bis-indole analogs possess a broad range of pharmacological properties, including anti-cancer, anti-bacterial, and anti-parasite properties [63,64]. A type of bis-indole alkaloid, hamacanthin, isolated from the sponge *Spongosorites* sp. demonstrated powerful antibacterial activity against methicillin-resistant *Staphylococcus aureus* [65]. In addition to several antileishmanial scaffolds reported, indole alkaloids showed promising activity against Leishmania parasites [66,67]. Docking studies have shown that bisindole analogs are potent inhibitors of pteridine reductase [64]. Here we found that dormant microsporidia spores can be inactivated in by MMV1593539, a bis-indole derivative. We show that treatment of spores with MMV1593539 causes both a decrease in viability and a decrease in sporoplasm invasion. Interestingly, MMV1593539 has been reported to have anthelmintic activity against the parasitic nematode *Haemonchus contortus*, but this compound is not active against *C. elegans* [45].

Microsporidia infect host cells through spore germination, and this process may be regulated by receptor proteins on cell membranes and external signals [68,69]. Furthermore, the changes in calcium ion concentrations, osmotic pressure of the external medium, or *in vivo* host environments can induce microsporidia spore firing [70–72]. However, many of the microsporidia proteins involved in spore firing are unknown. The subtilisin-like protease NbSLP1 has been implicated in germination as the active version localizes to the site of the spore where polar tube firing occurs in *N. bombycis* spores [73,74]. We previously found that protease inhibitors and quinones can inhibit spore firing [43]. Here we show that MMV1006203, a flavone, can prevent microsporidia invasion and spore firing. This inhibition occurs after incubation of the compound with spores, followed by washing of the spores, suggesting that this inhibitor acts directly on the spores to prevent firing. The molecular structures of displurigen and flavone are similar to that of MMV1006203, and these compounds also display the ability to prevent *N. parisii* infection. Other flavone compounds such as quercetin were shown to inhibit *Encephalitozoon intestinalis* infection, but it was not determined if these compounds block microsporidia invasion [75]. MMV1006203 was first shown to inhibit the human cytomegalovirus (HCMV) protease and more recently to have activity against *Plasmodium falciparum* [46,76].

Further work will be necessary to determine the molecular target of MMV1006203 in preventing microsporidia spore firing. Several approaches have been used in other eukaryotic intracellular parasites to identify targets of inhibitors. One approach is evolving strains that are resistant to an inhibitor and sequencing isolates to determine the genetic variants responsible for causing resistance [77]. A complementary biochemical-based approach is to use thermal

proteome profiling to identify proteins which have a change in thermal stability upon addition of the inhibitor. The efficacy and specificity of the compounds we have identified could also be further optimized by characterizing a collection of closely related analogs, which has been done for fumagillin [78].

## Supporting information

**S1 Data. All experimental data from this study.**
(XLSX)

## Acknowledgments

We are grateful to Winnie Zhao, Yin Chen Wan, Meng Xiao, Jonathan Tersigni, Hala Tamim El Jarkass, and Edward James for providing helpful comments on the manuscript. We thank the Medicines for Malaria Venture for providing the PRB compound library and the individual compounds to retest.

## Author Contributions

**Conceptualization:** Qingyuan Huang, Guoqing Pan, Aaron W. Reinke.

**Funding acquisition:** Aaron W. Reinke.

**Investigation:** Qingyuan Huang.

**Methodology:** Qingyuan Huang.

**Supervision:** Jie Chen, Guoqing Pan, Aaron W. Reinke.

**Visualization:** Qingyuan Huang.

**Writing – original draft:** Qingyuan Huang, Aaron W. Reinke.

**Writing – review & editing:** Qingyuan Huang, Jie Chen, Guoqing Pan, Aaron W. Reinke.

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
