## [Decision Letter · Decision Letter 0]

3 Oct 2023

Dear %TITLE% Reinke,

Thank you very much for submitting your manuscript "Screening of the Pandemic Response Box identifies anti-microsporidia compounds" for consideration at PLOS Neglected Tropical Diseases. The authors are to be congratulated for a strong manuscript that advances the field! 

As with all papers reviewed by the journal, your manuscript was reviewed by members of the editorial board and by several independent reviewers. The reviewers appreciated the attention to an important topic. Based on the reviews, we are likely to accept this manuscript for publication, providing that you modify the manuscript according to the review recommendations. 

Sincerely,

Joshua Nosanchuk, MD

Section Editor

Reviewer's Responses to Questions

**Key Review Criteria Required for Acceptance?**

**Methods**

-Are the objectives of the study clearly articulated with a clear testable hypothesis stated?

-Is the study design appropriate to address the stated objectives?

-Is the population clearly described and appropriate for the hypothesis being tested?

-Is the sample size sufficient to ensure adequate power to address the hypothesis being tested?

-Were correct statistical analysis used to support conclusions?

-Are there concerns about ethical or regulatory requirements being met?

Reviewer #1: (No Response)

Reviewer #2: This is a very nice study that demonstrates the utility of the C. elegans N. parisii system for moderate throughput screening of a drug library for compounds with activity against microsporidia. The methods are well documented and the statistical analysis is sound with sufficient sample size for determination of active compounds

**Results**

-Does the analysis presented match the analysis plan?

-Are the results clearly and completely presented?

-Are the figures (Tables, Images) of sufficient quality for clarity?

Reviewer #1: (No Response)

Reviewer #2: The analysis of the data is well presented and the figures clearly provide the data set needed for interpretation of the experiments. The authors found a benznidazole compound related to albendazole as one active drug. This makes sense as the literature supports albendazole (a benznidazole tubulin binder) as a proven therapeutic agent for several microsporidia. The other class of compounds with activity were flavones, for which the mechanism(s) by which germination is inhibited is not known.

**Conclusions**

-Are the conclusions supported by the data presented?

-Are the limitations of analysis clearly described?

-Do the authors discuss how these data can be helpful to advance our understanding of the topic under study?

-Is public health relevance addressed?

Reviewer #1: (No Response)

Reviewer #2: The conclusions are supported by the data and the discussion is appropriate for the data presented. There is a need for new therapeutic agents for these pathogens so the paper does address an active public health issue.

**Editorial and Data Presentation Modifications?**

Reviewer #1: (No Response)

Reviewer #2: I would suggest at line 82 when Pancytospora is introduced the authors should indicate it is related to Enterocytozoon and Vittaforma rather than just staying "it is related to human-infecting species"

**Summary and General Comments**

Reviewer #1: Introduction:

Microsporidia are obligate intracellular parasites that cause disease in many commercially important invertebrates as well as humans in some limited settings. However, few treatment options exist for controlling microsporidia infections in any organism. The authors search for potential microsporidia therapeutic agents using a screen they previously designed and implemented in the nematode Caenorhabditis elegans. The screen measures the beneficial effects of compounds on the fecundity of microsporidia-infected worms. While this design may result in missing agents that kill microsporidia but are toxic to worms, it also removes compounds that may be unsuitable for use due to detrimental effects on the host. Another advantage of the screen is that it has the potential to find compounds that work by reducing pathogen loads by acting at various stages of the lifecycle and those that work by increasing tolerance to infection. Previously the authors tested 2560 FDA-approved compounds and natural products, and identify 11 candidate microsporidia inhibitors (Murareanu et al., 2022). In the submitted manuscript, the authors test a smaller set of novel compounds (400) from a collection called the Pandemic Response Box, and initially identify 4 novel inhibitors before finding additional promising candidates by exploring compounds that are chemically related to the original ‘hit’ compounds. The diminished novelty and scope of the screen reduces the impact of this manuscript relative to the previous report. However, this is important work that few groups are engaged in and the manuscript details some very interesting findings. As such, I would only ask that they add a few discussion points to their manuscript. First, the authors state that MMV1782387 works by decreasing proliferation (occurring in the meront through sporoblast stages). It appears that they also find that this compound reduces sporoplasm number (without an impact on firing) (Figure 4 and comments in discussion). They cleverly remove this effect from consideration in the anti-proliferation assay by only adding drug after invasion, but I think this point and their work-around are worth emphasizing. Two other compounds (MMV1593539 and MMV1634491) appear to reduce sporoplasm number (without any impact on proliferation or spore firing). As 3 compounds (MMV1593539, MMV1634491, MMV1782387) seem to have an impact on sporoplasm number / invasion (but not spore firing), could the authors comment on a potential mechanism for this 'invasion' level effect? Second, I was struck by the differential anti-microsporidia and host toxicity effects of the various benzimidazole compounds. Could the authors discuss the likely avenue of host toxicity with these compounds? Is it host beta tubulin? And if so what’s known about the structure / function of these compounds and both parasite and host beta tubulin that might explain the different sensitivities of host and pathogen? Finally, I was curious to know what the authors see as the future steps for advancing these and previously identified compounds into a more applied / translational space?

A small point is that the statements about the beta-tubulin amino acids that potentially mediate sensitivity or resistance to albendazole (line 214-218) should be moved to the discussion.

References

Murareanu, B. M., Antao, N. V., Zhao, W., Dubuffet, A., Alaoui, H. E., Knox, J., Ekiert, D. C., Bhabha, G., Roy, P. J. and Reinke, A. W. (2022). High-throughput small molecule screen identifies inhibitors of microsporidia invasion and proliferation in C. elegans. Nat. Commun. 13, 5653.

Reviewer #2: Overall, this is an excellent paper with a useful moderate throughput screening system. While screening compounds is not novel, this is important work and the compounds identified have the potential to move forward as therapeutic leads. In particular, no significant work has been done on flavones as a therapeutic class for these pathogens and this is a novel area of research suggested by this paper.

PLOS authors have the option to publish the peer review history of their article (what does this mean?). If published, this will include your full peer review and any attached files.

Reviewer #1: No

Reviewer #2: Yes: Louis M. Weiss

Figure Files:

Data Requirements:

Reproducibility:

References

---

## [Decision Letter · Decision Letter 1]

20 Nov 2023

Dear %TITLE% Reinke,

Thank you for your thoughtful response to the reviewers' comments. We are pleased to inform you that your manuscript 'Screening of the Pandemic Response Box identifies anti-microsporidia compounds' has been provisionally accepted for publication in PLOS Neglected Tropical Diseases.

Best regards,

Joshua Nosanchuk, MD

Section Editor

Joshua Nosanchuk

Section Editor

Reviewer's Responses to Questions

**Key Review Criteria Required for Acceptance?**

**Methods**

-Are the objectives of the study clearly articulated with a clear testable hypothesis stated?

-Is the study design appropriate to address the stated objectives?

-Is the population clearly described and appropriate for the hypothesis being tested?

-Is the sample size sufficient to ensure adequate power to address the hypothesis being tested?

-Were correct statistical analysis used to support conclusions?

-Are there concerns about ethical or regulatory requirements being met?

Reviewer #1: Good.

**Results**

-Does the analysis presented match the analysis plan?

-Are the results clearly and completely presented?

-Are the figures (Tables, Images) of sufficient quality for clarity?

Reviewer #1: Good.

**Conclusions**

-Are the conclusions supported by the data presented?

-Are the limitations of analysis clearly described?

-Do the authors discuss how these data can be helpful to advance our understanding of the topic under study?

-Is public health relevance addressed?

Reviewer #1: Good.

**Editorial and Data Presentation Modifications?**

Reviewer #1: (No Response)

**Summary and General Comments**

Reviewer #1: The authors have fully addressed my minor suggestions and is suitable for publication in its present form.

PLOS authors have the option to publish the peer review history of their article (what does this mean?). If published, this will include your full peer review and any attached files.

Reviewer #1: No

---

## [Editor Report · Acceptance letter]

3 Dec 2023

Dear Reinke,

We are delighted to inform you that your manuscript, "Screening of the Pandemic Response Box identifies anti-microsporidia compounds," has been formally accepted for publication in PLOS Neglected Tropical Diseases.

Best regards,

Shaden Kamhawi

co-Editor-in-Chief

Paul Brindley

co-Editor-in-Chief
